# Lightweight Multi-Class Support Vector Machine-Based Medical Diagnosis System with Privacy Preservation

**DOI:** 10.3390/s23229033

**Published:** 2023-11-08

**Authors:** Sherif Abdelfattah, Mohamed Baza, Mohamed Mahmoud, Mostafa M. Fouda, Khalid Abualsaud, Elias Yaacoub, Maazen Alsabaan, Mohsen Guizani

**Affiliations:** 1Department of Computer Science and Information Systems, Bradley University, Peoria, IL 61625, USA; sabdelfattah@bradley.edu; 2Department of Computer Science, College of Charleston, Charleston, SC 29424, USA; bazam@cofc.edu; 3Department of Electrical and Computer Engineering, Tennessee Technological University, Cookeville, TN 38505, USA; mmahmoud@tntech.edu; 4Department of Electrical and Computer Engineering, College of Science and Engineering, Idaho State University, Pocatello, ID 83209, USA; mfouda@ieee.org; 5Center for Advanced Energy Studies (CAES), Idaho Falls, ID 83401, USA; 6Department of Computer Science and Engineering, Qatar University, Doha 2713, Qatar; eliasy@ieee.org; 7Department of Computer Engineering, College of Computer and Information Sciences, King Saud University, Riyadh 11451, Saudi Arabia; malsabaan@ksu.edu.sa; 8Machine Learning Department, Mohamed bin Zayed University of Artificial Intelligence, Abu Dhabi P.O. Box 131818, United Arab Emirates; mguizani@ieee.org

**Keywords:** privacy preservation, cloud security, medical diagnosis, support vector machine (SVM), multiclassification

## Abstract

Machine learning, powered by cloud servers, has found application in medical diagnosis, enhancing the capabilities of smart healthcare services. Research literature demonstrates that the support vector machine (SVM) consistently demonstrates remarkable accuracy in medical diagnosis. Nonetheless, safeguarding patients’ health data privacy and preserving the intellectual property of diagnosis models is of paramount importance. This concern arises from the common practice of outsourcing these models to third-party cloud servers that may not be entirely trustworthy. Few studies in the literature have delved into addressing these issues within SVM-based diagnosis systems. These studies, however, typically demand substantial communication and computational resources and may fail to conceal classification results and protect model intellectual property. This paper aims to tackle these limitations within a multi-class SVM medical diagnosis system. To achieve this, we have introduced modifications to an inner product encryption cryptosystem and incorporated it into our medical diagnosis framework. Notably, our cryptosystem proves to be more efficient than the Paillier and multi-party computation cryptography methods employed in previous research. Although we focus on a medical application in this paper, our approach can also be used for other applications that need the evaluation of machine learning models in a privacy-preserving way such as electricity theft detection in the smart grid, electric vehicle charging coordination, and vehicular social networks. To assess the performance and security of our approach, we conducted comprehensive analyses and experiments. Our findings demonstrate that our proposed method successfully fulfills our security and privacy objectives while maintaining high classification accuracy and minimizing communication and computational overhead.

## 1. Introduction

Recent advancements in artificial intelligence (AI) and computational technologies have played a pivotal role in shaping the concept of smart healthcare systems [1]. These cutting-edge systems leverage the capabilities of AI algorithms to swiftly and accurately process medical data, empowering healthcare providers to make well-informed decisions and provide personalized patient care. The primary objective of these systems is to make healthcare more accessible and cost-effective, particularly in response to the challenges posed by aging populations and a shortage of medical professionals. Moreover, by offering remote healthcare services, they bridge the gap in medical care access for remote and underserved rural areas. The significance of healthcare systems is underscored by substantial investments, such as the estimated $22 billion in telemedicine services in the USA [2]. At the core of these smart healthcare systems lies machine learning (ML)-based remote medical diagnosis, which can both identify existing diseases and predict and prevent future ones. One compelling example of their importance is the ML system developed by Google, capable of detecting malignant tumors through the analysis of mammograms [3].

Among other ML techniques, SVM stands out for its high accuracy and efficiency in the context of medical applications, which frequently involve small datasets [4,5,6]. Like other ML methods, SVM follows a two-phase process: training and testing. During the training phase, healthcare providers employ historical medical data from patients to create an SVM model. However, due to resource constraints and the need for uninterrupted diagnosis services, healthcare providers often outsource this model to a third-party cloud server operated by an independent entity. In the testing phase, users like doctors and patients submit medical data, including vital signs and symptoms, to the cloud server. The cloud server utilizes this data to evaluate the model and subsequently provides the classification or diagnosis to the users [7].

While this cloud-based approach enables medical diagnosis services for individuals and healthcare centers, the inherent trust issues associated with third-party cloud servers necessitate safeguarding patient data confidentiality and the intellectual property of the diagnosis model. Therefore, it is imperative to explore a methodology that allows the cloud server to receive and evaluate data without learning the data, diagnosis, or model parameters. This can be achieved through encryption of both the model and the data, enabling the cloud server to use the encrypted model and encrypted data to compute the diagnosis in ciphertext format, decipherable solely by the user who submitted the data.

In the existing literature, only a limited number of studies have endeavored to address the issues of privacy and safeguarding the intellectual property of models in SVM-based medical diagnosis systems [8,9,10,11,12]. However, these studies have notable shortcomings. Some, such as those in [8,9,10], primarily focus on privacy concerns but do not consider hiding the model’s parameters. Conversely, others, like [11,12], enable the server to learn the classification (or diagnosis) results which creates a vulnerability that can be used to breach the privacy of the users. Furthermore, existing approaches predominantly rely on cryptosystems that demand substantial communication and computational resources such as multi-party computations and Pailler cryptography. Other approaches also involve users in the computation of classifications, which is not practical given that users typically possess limited computational capabilities. Among these studies, the works in [12,13] are the most pertinent to our research.

The approach presented in [12] exhibits various drawbacks. It relies on an Okamoto–Uchiyama (OU) cryptosystem [12] based on Paillier cryptography [14], necessitating extensive computational resources. Additionally, as the number of classes increases, the computational demands on model users escalate significantly. Furthermore, their proposed method mandates the active involvement of the model owner in every classification operation, which detracts from the advantages of outsourcing the model to a cloud server, as it requires continuous online interaction and allocation of computational resources. Another limitation is that to update the model’s parameters, ref. [12] necessitates transmitting updated encrypted model parameters to each model user, resulting in a substantial communication burden. While the study presented in [13] has tackled several limitations, including safeguarding the model’s intellectual property, preserving user data privacy, and concealing classification results, it does so by relying on a setup involving two non-colluding servers. Each server has the capability of performing specific computations but possesses limited knowledge. However, implementing such a dual-server setup can be expensive and may pose practical challenges. Additionally, the proposed approach incurs substantial communication and computational resources, particularly as more classes (i.e., diseases) are considered by the model, owing to its utilization of multi-party computation and Paillier cryptography.

This paper endeavors to overcome the aforementioned limitations by introducing an efficient multi-class SVM-based medical diagnosis system that preserves privacy and safeguards model intellectual property. In contrast to existing methods reliant on inefficient cryptography, our approach centers on an inner product encryption cryptosystem, characterized by efficient computation and communication. Our proposed method ensures the privacy of the medical health of users by not allowing anyone to obtain their data and diagnosis results. Additionally, our approach aims to protect the intellectual property by not allowing anyone to obtain the plaintext parameters of the model. Nevertheless, the server can still employ the ciphertext of the users’ data and encrypted model parameters to obtain a masked classification (or diagnosis). In our approach, we have modified the inner product encryption cryptosystem introduced by Kim et al. [15] to enhance its efficiency within our context. Specifically, the cryptosystem outlined in [15] was originally designed for a scenario involving a single encryptor and a single decryptor, where the encryptor serves as the model owner, and the decryptor acts as a model user. However, this cryptosystem necessitates outsourcing an encrypted model to the cloud server for each user and sharing a key between each user and the model owner, a process that is notably inefficient and unscalable. To address this, we have modified the cryptosystem in [15] to make it suitable for a single-encryptor and multiple-decryptors setting. Consequently, only one encrypted model needs to be outsourced to the cloud, which can then be utilized by all users efficiently. To further optimize the efficiency of our approach, we diverge from the conventional privacy-preserving SVM schemes, as seen in many existing works [11,16,17,18], which are designed primarily for binary classification tasks. Instead, we develop a multi-class SVM classifier capable of diagnosing multiple diseases that share similar symptoms. Our method employs multiple binary classifiers rather than a single multi-class classifier. This necessitates encrypting and outsourcing each individual model to the cloud server, resulting in multiple computations required by the server to evaluate these models. While our paper primarily concentrates on medical diagnosis, the methodology we propose is versatile and applicable to various other domains, including the detection of electricity theft and the coordination of electric vehicle charging.

A real medical dataset in [19] is used to train our SVM diagnosis model for dermatology diseases. This dataset is extensively utilized in the literature [12,20] and widely regarded as a benchmark in the field of medical applications. The dataset has six classes (skin diseases), including pityriasis rubra pilaris, psoriasis, lichen planus, pityriasis rosea, chronic dermatitis, and seboreic dermatitis. The dataset has 34 features, including age, family history, and several symptoms. To assess our proposal, extensive evaluations are conducted, and the classification accuracy and computation and computation overhead are the main metrics measured. The main contributions of this paper can be summarized as follows:A privacy-preserving and lightweight SVM medical diagnosis scheme has been proposed by modifying an inner product encryption cryptosystem.The results of our analysis demonstrate that our proposed scheme successfully fulfills our security and privacy objectives, including preserving the privacy of the patient’s health status and protecting the model’s parameters.Our evaluations indicate that our proposal outperforms the most relevant approaches in the overhead while maintaining high classification accuracy.

This paper is derived from the PhD thesis of the first author [21]. The structure of the remaining sections in this paper is as follows. The main entities considered in our system and the main messages exchanged among them, in addition to the objectives of the attackers, are discussed in Section 2. Section 3 offers an overview of the fundamental concepts used in our paper. Section 4 provides an in-depth examination of our proposed scheme. We assess the capability of our scheme to preserve privacy in Section 5. Section 6 presents the results of experiments conducted to assess communication/computation overhead and model accuracy. Section 7 offers insights into related works. Lastly, our conclusions are summarized in Section 8.

## 2. Network and Threat Models

This section first explains the network model considered in this paper, including a description of the main entities and their communications. After that, we explain the threat model describing the entities that could be attackers and their capabilities and objectives.

### 2.1. Network Model

As illustrated in Figure 1, the network model includes four main entities, including a model owner (*MO*), users (*MU*s), a cloud server (*CS*), and an offline key distribution center (*KDC*). In this subsection, we delve into the specific functions of each of these entities and elucidate the nature of their inter-communications. This elucidation aims to offer a comprehensive understanding of how these integral components collaborate within the proposed framework, fostering clarity and insight into the scheme’s operational dynamics.

*KDC*. The key distribution center is a trusted entity that is responsible for computing secret keys needed to execute our scheme and distribute them to the other entities in the system. To fulfill the trustworthiness of this entity, it can be implemented by a trusted party, e.g., the Department of Health. Although the detailed implementation of the key distribution center is out of this paper’s scope, in the literature, there are many proposed approaches that can implement the *KDC* without the need for a trusted entity.*MO*. The model owner is a healthcare center that owns medical datasets for patients and uses them to compute a diagnosis model. It is reasonable to assume that the dataset is small. This is an acceptable assumption in medical diagnosis. The model owner does not have enough computation resources to provide the diagnosis service, so it outsources the diagnosis model after encrypting it to a third-party cloud service. Then, users send their data to the cloud to classify them using the model. Using a third-party cloud service can provide several benefits such as no computation burden being needed from the model owner and also the medical diagnosis service is always available.*MUs*. The model users include any party that seeks the medical diagnosis service, such as patients, doctors, clinics, healthcare monitoring services, etc. As indicated in Figure 1, users send encrypted medical data that includes vital and symptom data to the cloud server, which uses the ciphertext to compute encrypted classification and send it to the user to decrypt.*CS*: The cloud server offers an online medical diagnosis service. It is possessed and run by a third party, which can be a private company. It uses the ciphertext of the medical data sent by users and the ciphertext of the diagnosis model outsourced by the model owner to conduct operations over encrypted data. The result of these operations is the ciphertext of the classification (or medical diagnosis). Note that the cloud server cannot decrypt any of these ciphertexts to achieve our security and privacy objectives.

### 2.2. Threat Model

To achieve high security and privacy protection for our system, we consider a large spectrum of possible attackers, including model users, external eavesdroppers, and the cloud server. We assume that attackers have the capability to collect all messages exchanged in the system, and then analyze them to infer sensitive information. The attackers are honest-but-curious in the sense that they do not want to make a disruption or cause a malfunction to the system but they want to either collect sensitive information on the patient’s medical data [22] or steal the model by learning its plaintext parameters. Note that the intellectual property of the model is owned by the model owner, and thus no one else should be able to obtain it.

Therefore, to achieve our security and privacy objectives, our model should be secure against several attack models, including *Known-background*, *Known-plaintext model*, and *Known-ciphertext*. In *Known-background model*, attackers aim to infer patient classifications by using some background information, such as the probability of each classification and statistical data on the model’s medical classifications. Other attackers may not be interested in the individual patient classifications but they want to collect statistical information on the spread of a certain disease [22,23]. For the *Known-plaintext model*, it is assumed that attackers can gather pairs of plaintext and corresponding ciphertext messages exchanged in the system and then analyze these pairs to infer any information that can be used to figure out new patients’ medical data [24]. In *Known-ciphertext model*, we assume that attackers can gather ciphertext messages exchanged in the system without being able to know their plaintext messages. The attackers analyze the ciphertext messages trying to infer information on the encryption scheme that can help compute the plaintext of the new ciphertext messages [25,26].

## 3. Preliminaries

In this section, we provide a concise overview of the bilinear pairing, support vector machine, and a function-hiding inner-product encryption scheme.

### 3.1. Bilinear Pairing

Let *m* denote the prime order of the cyclic groups G and $G_{T}$, with the generator of G being *V*. We assume that G and $G_{T}$ possess a non-degenerate and efficiently computed bilinear pairing map ($\hat{e}$):$$\hat{e}:G\times G\to G_{T}$$
satisfying the following properties:$\hat{e}\left(V,V\right)\neq 1_{G_{T}}$$\hat{e}\left(xV_{1},yS_{1}\right)=\hat{e}{\left(V_{1},S_{1}\right)}^{xy}\in G_{T}$, for all $x,y\in Z_{m}^{*}$ and any $V_{1},S_{1}\in G$, where $Z_{m}^{*}$ represents a finite field of order *m*.

### 3.2. Support Vector Machine (SVM)

SVMs have garnered extensive recognition and application within the field of medical classification, primarily owing to their remarkable accuracy [27,28,29]. Various notable applications within the medical domain have harnessed SVM’s potential, demonstrating its efficacy. These encompass diverse tasks, such as the classification of electrocardiogram signals [30], aiding in clinical diagnosis [31,32], identifying individuals on the autism spectrum [33], and detecting cervical cancer cells [34], among numerous others. SVM’s versatility and precision make it a valuable tool for addressing critical medical challenges.

#### 3.2.1. Linear SVM

Linear support vector machine (LSVM) is a powerful tool primarily used for binary classification tasks. An illustration of the linear SVM is shown in Figure 2. In this figure, the blue circles are used to represent the benign data while the green triangles are used to represent the malicious data. As depicted in Figure 2, the core objective of LSVM is to discern an optimal hyperplane within the feature space, capable of effectively segregating the given set of training samples into two distinct classes. This hyperplane is essentially defined by a *decision function*. Let us suppose we have a training dataset comprising *m* samples, denoted as $\left\{\left(t_{1},y_{1}\right),\ldots ,\left(t_{m},y_{m}\right)\right\}$, where each $t_{j}$ belongs to the real-valued vector space $R^{n}$, representing the *j*th sample with *n* elements. The corresponding $y_{j}$ is a label assigned to each sample, taking values from the set $\left\{-1,+1\right\}$. Typically, −1 corresponds to one class, while +1 signifies the other class.

The predictions and classifications for new, unseen data can be made after completing the training phase using the decision function defined in Equation (Equation 1):(1)$$d\left(t\right):=w\cdot t+b=\sum_{s\in S}\alpha_{s}y_{s}x_{s}\cdot t+b$$

Here, w, t, and *b* denote the weight vector, the unlabeled sample’s vector of features, and the bias, respectively. *S* represents the set of support vectors $\alpha_{s}$, $x_{s}$, and $y_{s}$ stands for the Lagrange multiplier of the support vector, the support vector, and its associated class label, respectively. The classification of an unlabeled sample *t* can be determined by using Equation (Equation 2) to compute d(t), and then, the classification is benign when is negative; otherwise, it is malicious.
(2)$$\begin{array}{c} \mathrm{classification}\;\mathrm{result}=\left\{\begin{array}{cc} +ve, & \mathrm{if}\;d(t)>0\\ -ve, & \mathrm{if}\;d(t)<0 \end{array}\right. \end{array}$$

#### 3.2.2. Multi-Classification SVM (MCSVM)

Constructing an MCSVM classifier typically involves creating multiple two-class classifiers. Figure 3 illustrates two conventional techniques for building multi-class classifiers using SVM: the one-versus-all (1VA) and one-versus-one (1V1) approaches. Four classes are used in the figure.

In the 1V1 approach, a hyperplane is computed between every pair of classes, resulting in a *k*-class classifier. Consequently, $C_{k}^{2}=\frac{k(k-1)}{2}$ is the total number of hyperplanes. In the example shown in Figure 3, the 1V1 approach establishes six binary classifiers. On the other hand, the 1VA approach involves computing a hyperplane between every class and the remaining classes. In this case, only *k* hyperplanes are needed. In the example shown in Figure 3, four binary classifiers are established by the 1VA approach. Obviously, compared to the 1V1 approach, the 1VA approach creates fewer classifiers.

Due to its efficiency, we opt for the 1VA approach. To implement this method, we train $N_{c}$ SVM models. During the training phase, each classifier uses class *j* as the +*ve* class and the remaining samples as the −*ve* class. The decision function of the MCSVM classifier during the testing phase, denoted as d(t), where t is a given normalized test sample, can be calculated as follows:(3)$$d\left(t\right)=\underset{j=1,\ldots ,N_{c}}{\mathrm{argmax}}\left(d_{j}\left(t\right)\right)$$
where $d_{j}\left(t\right)$ represents the decision function of the *j*th classifier and can be computed as follows:(4)$$d_{j}\left(t\right):=w_{j}\cdot t+b_{j}$$

### 3.3. Function-Hiding Inner-Product Encryption Cryptosystem

The research in [15] introduced a functional encryption method enabling the computation of the inner product of two vectors using their ciphertexts without revealing the vectors themselves. In this scheme, the inner product (x·y) can be calculated, where x and y are the ciphertexts of vectors. This inner product encryption system is classified as function-hiding since it discloses no additional information about x and y other than the result of the inner product.

Choose $g_{1}$ as generators for the multiplicative group $G_{1}$, and $g_{2}$ as generators for the multiplicative group $G_{2}$. Let the two groups be of prime order *q*. Define the bilinear pairing map $\hat{e}:G_{1}\times G_{2}\to G_{T}$, which maps elements from $G_{1}$ and $G_{2}$ to elements in the target group $G_{T}$ of prime order *q*. Consider a polynomial-sized subset *S* of $Z_{q}$. Define $B^{\prime }=det\left(B\right)\cdot {\left(B^{-1}\right)}^{T}$, where $B\overset{}{\leftarrow }{\mathrm{GL}}_{n}\left(Z_{q}\right)$. Here, ${\mathrm{GL}}_{n}\left(Z_{q}\right)$ represents the general linear group of $\left(n\times n\right)$ matrices over $\left(Z_{q}\right)$.

The public key and the master secret key are denoted as pk and msk, respectively, where $pk=(G_{1},G_{2},G_{T},g_{1},g_{2},q,\hat{e},S)$, and msk=(B and $B^{\prime })$. The encryption scheme involves three phases as follows:KeyGen(msk,x): Given msk and a vector $x\in Z_{q}^{n}$, the algorithm selects a uniformly random element $\alpha \overset{R}{\leftarrow }Z_{q}$ and produces the secret key pair $sk=\left(K_{1},K_{2}\right)=\left(g_{1}^{\alpha \cdot det(B)},g_{1}^{\alpha \cdot x\cdot B}\right)$, where sk and $K_{2}$ are the secret key and *n* elements vector that has values from $G_{1}$, respectively.Encrypt(msk,y): Given msk and a vector $y\in Z_{q}^{n}$, the algorithm selects a uniformly random element $\beta \overset{R}{\leftarrow }Z_{q}$ and produces $ct=\left(C_{1},C_{2}\right)=\left(g_{2}^{\beta },g_{2}^{\beta \cdot y\cdot B^{\prime }}\right)$, where ct and $C_{2}$ are the ciphertext pair and *n* element vector that has values from $G_{2}$, respectively.Decrypt(pk,sk,ct): Given pk, sk, and ct, the algorithm computes the inner product x·y. The decryption algorithm produces $D_{1}=\hat{e}\left(K_{1},C_{1}\right)=\hat{e}{\left(g_{1},g_{2}\right)}^{\alpha \beta \cdot det(B)}$ and $D_{2}=\hat{e}\left(K_{2},C_{2}\right)=\hat{e}{\left(g_{1},g_{2}\right)}^{\alpha \beta \cdot det(B)z}$, where z=x·y. The algorithm then checks whether there exists *z* such that ${\left(D_{1}\right)}^{z}=D_{2}$ by computing a discrete logarithm in $G_{T}$ using methods such as the baby-step giant-step algorithm [35]. If a valid *z* is found, it is output; otherwise, ⊥ is output to indicate that no valid *z* exists.

This encryption scheme is originally designed for a single encryptor, which is *MO* in our scheme, and a single decryptor, which is *MU* in our scheme. The encryptor must share a unique key pair (B and $B^{\prime })$ with every decryptor. If this method is employed in our research, the model parameters would need to be encrypted *n* times, where *n* represents the number of MOs. This results in significant computation and communication overhead, which is not feasible. In this research paper, we have adapted this scheme to accommodate a scenario with a single model owner and multiple model users. In this setup, each model user and the model owner utilize unique keys for their operations. In this modified scheme, even though the model parameters are encrypted using the model owner’s specific key, the inner product computation can be performed when the medical data are encrypted using any of the model users’ keys. This approach significantly reduces the computation and communication overhead, as the model parameters need to be encrypted only once, regardless of the number of model users.

## 4. Proposed Scheme

In this section, the term “hospital”, denoted as H, signifies the entity owning the model, while “patient”, represented by P, refers to the user of the model. The proposed methodology unfolds through four distinct phases. During the *system initialization* stage, the key distribution center (*KDC*) undertakes the computation and distribution of secret keys, disseminating them to both H and P. Moving on to the *model encryption* phase, each support vector machine (SVM) model’s parameter vector is encrypted by H. Subsequently, all these encrypted parameters, along with random numbers employed for masking classification results, are outsourced to the cloud server (*CS*). Transitioning to the *medical data encryption* step, P encrypts their medical data vector, including symptoms and vital data. The encrypted data are then sent to the server to input to the diagnosis model and compute the masked classification (or diagnosis) score in the *medical diagnosis* step. These masked classifications are then transmitted back to P for the process of unmasking and subsequent understanding.

### 4.1. Design Objectives

We aim to accomplish specific design objectives in our approach:*Preserving Privacy*. In our proposed scheme, it is imperative to protect the confidentiality of the model user’s medical data. The outsourced medical data of the *MU* must remain confidential, ensuring that neither the cloud server (*CS*) nor the model owner (*MO*) can access any information about it. Additionally, the classification results should remain concealed from the *CS*, with only the *MU* having access to this information.*Protecting Intellectual Property*. Our scheme is designed to preserve the confidentiality of the diagnosis model’s parameters from potential threats posed by the *CS*, *MU*, and external eavesdroppers.*High Diagnosis Accuracy and Low Communication/Computation Cost*. The proposed scheme aims to deliver precise medical diagnoses for *MU*s while minimizing computational and communication burdens. Considering the limited computational and communication resources typically available to *MU*s, it is crucial to optimize their involvement in the online diagnosis process. The encryption methods utilized for user requests should be lightweight, allowing data users to remain offline during the online diagnosis process until they receive the classification results. Moreover, heavy computation and communication tasks should be offloaded to the *CS*, which possesses ample communication and computation resources. Notably, the cloud server should be able to compute the medical diagnosis without the need for the participation of the model owner because requiring the model owner to be online and interactive all the time to help the server in the diagnosis computations diminishes the benefits of outsourcing the medical diagnosis to a third party.

### 4.2. System Initialization

The following algorithms are used by the key distribution center (*KDC*) to generate the secret keys of the patients and hospitals.

$\mathrm{InitializeSetup}\left(1^{\lambda }\right)\overset{}{\to }\mathrm{PP},\mathrm{MSK}$. This algorithm takes $1^{\lambda }$ as the security parameter input, where λ represents the security parameter, and produces PP and MSK, which represent the public parameters and the master secret key, respectively.

The key distribution center (*KDC*) generates the master secret key $\mathrm{MSK}=\left\{M,N_{1},N_{2}\right\}$ randomly from the set of invertible matrices ${\mathrm{GL}}_{m+2}\left(Z_{q}\right)$, where $\left\{M,N_{1},N_{2}\right\}$ are matrices of dimensions (m+2)×(m+2). Here, *m* represents the size of the patient’s medical data, and ${\mathrm{GL}}_{m+2}\left(Z_{q}\right)$ denotes the general linear group of (m+2)×(m+2) matrices over the field $Z_{q}$. Subsequently, the output of this process yields the public parameters $\mathrm{PP}=(G_{1},G_{2},G_{T},g_{1},g_{2},q,e,S)$.

$\mathrm{GenerateHospitalKey}\left(\mathrm{MSK}\right)\overset{}{\to }\mathrm{HSK}$. For hospital H, the algorithm yields secret key HSK, crucial for encrypting the parameters of each SVM binary classifier. The computation is outlined as follows:$$\begin{array}{cc} \mathrm{HSK} & =\{MN_{1},\;MN_{2}\} \end{array}$$

Subsequently, *KDC* sends HSK to the H.

$\mathrm{GeneratePatientKey}\left(\mathrm{MSK}\right)\overset{}{\to }{\mathrm{PSK}}_{P}$. The secret key ${\mathrm{PSK}}_{P}$ is computed according to the following scheme for every patient P:$$\begin{array}{cc} {\mathrm{PSK}}_{P} & =\{N_{1}^{-1}A_{p},\;N_{2}^{-1}B_{p}\} \end{array}$$
where $\left\{A_{p},B_{p}\right\}$ represents $\left(m+2)\times (m+2\right)$ matrices of randomly generated values, satisfying $A_{p}+B_{p}=M^{-1}$. Subsequently, the patient P obtains ${\mathrm{PSK}}_{p}$ from the *KDC*. This key is utilized by the patient to encrypt their medical data, which are then transmitted to the cloud for medical diagnosis.

### 4.3. Model Encryption

To securely transmit the encrypted SVM models’ parameters to the cloud server, H performs the following procedure.

$\mathrm{SecureEncryptModel}\left(\mathrm{PP},\mathrm{HSK},W_{j}\right)\overset{}{\to }C_{W_{j}}$. This operation uses the public parameters PP, the hospital secret key HSK, and the model’s parameter vector $W_{j}$ as input and produces the encrypted SVM model parameters $C_{W_{j}}$.

The parameters of each model *j* are denoted as $W_{j}=\left\{w_{j,1},\cdots ,w_{j,m},b_{j}\right\}\in Z_{q}^{m+1}$, where j=1,⋯,k, and *k* represents the total number of the models. H then constructs the $W_{j}$ vector with (m+2)-element where the first m+1 elements will be filled the model parameters and places one at the following (m+2)−th element. In order to mask the classification, the (m+2)−th element will be multiplied by the corresponding element in the patient’s data vector, which stores the masking number. To encrypt $W_{j}$, the hospital selects a uniformly random element $\alpha \overset{R}{\leftarrow }Z_{q}$ and generates,
$$\begin{array}{cc} C_{W_{j}} & =\left\{C_{W_{j}}^{1},C_{W_{j}}^{2},C_{W_{j}}^{3}\right\}=\left\{g_{1}^{\alpha },g_{1}^{\alpha \cdot W_{j}MN_{1}},g_{1}^{\alpha \cdot W_{j}MN_{2}}\right\} \end{array}$$

Here, $C_{W_{j}}^{2}$ and $C_{W_{j}}^{3}$ are vectors of size (m+2)$G_{1}$ elements. H then transfers the encrypted model’s parameters ${\left\{C_{W_{j}}\right\}}_{j=1\cdots k}$ to the cloud server.

### 4.4. Encryption of Medical Data

During this stage, the subsequent algorithm is utilized to encrypt the medical data by P and transmits it to the cloud server for classification.

$\mathrm{MedicalDataEncryption}\left(\mathrm{PP},{\mathrm{PSK}}_{P},T_{P}\right)\overset{}{\to }C_{T_{P}}$. This algorithm takes the public parameters PP, the patient’s secret keys ${\mathrm{PSK}}_{P}$, and the medical data vector $T_{P}$ as inputs, producing the encrypted medical data $C_{T_{P}}$. The P’s medical data for the patient is denoted as $\left\{t_{1},\cdots ,t_{m},1\right\}\in Z_{q}^{m+1}$. The patient then constructs $T_{P}=\left\{t_{1},\cdots ,t_{m},1,b_{P}\right\}\in Z_{q}^{m+2}$ with m+2-elements, encompassing the medical data in the first m+1 elements, using the (m+2)−th element for storing the randomly generated masking number $b_{P}$.

After that, P selects a uniformly random element $\beta \overset{R}{\leftarrow }Z_{q}$ to encrypt $T_{P}$ as follows
$$\begin{array}{cc} C_{T_{P}} & =\left\{C_{T_{P}}^{1},C_{T_{P}}^{2},C_{T_{P}}^{3}\right\}=\left\{g_{2}^{\beta },g_{2}^{\beta \cdot N_{1}^{-1}A_{p}T_{P}^{⊤}},g_{2}^{\beta \cdot N_{2}^{-1}B_{p}T_{P}^{⊤}}\right\} \end{array}$$

It is noteworthy that $C_{T_{P}}^{2}$ and $C_{T_{P}}^{3}$ are arrays of size (m+2)$G_{2}$ elements. Ultimately, the patient sends $C_{T_{P}}$ to CS for the diagnosis.

### 4.5. Classification

Using the ciphertext of the model parameters and the ciphertext of the medical data of users, in this phase, the server computes the inner product result of the the model parameters and medical data which results in masked classification score (i.e., the medical diagnosis). Figure 4 outlines the computations performed by *CS* to classify the medical diagnosis. The specifics are as follows.

$\mathrm{Diagnosis}\left(\mathrm{PP},C_{W_{j}},C_{T_{P}}\right)\overset{}{\to }z_{j}$: The encrypted model’s parameters $C_{W_{j}}$ and the encrypted patient’s medical data $C_{T_{P}}$ are the algorithm’s input, which then produces the masked classification $z_{j}$.

A random positive number *r* is used by the cloud server for the model’s intellectual property protection. This protection can be accomplished by multiplying *r* with the first m+1 elements of $C_{W_{j}}$. $D_{1}=\hat{e}\left(C_{W_{j}}^{1},C_{T_{P}}^{1}\right)=\hat{e}\left(g_{1}^{\alpha },g_{2}^{\beta }\right)$ and $D_{2}={\left(E_{1}\times E_{2}\right)}^{r}$ are used to compute the inner product of $C_{W_{j}}$ and $C_{T_{P}}$ for every model *j*, where
$$\begin{array}{cc} E_{1} & =\hat{e}\left(C_{W_{j}}^{2},C_{T_{P}}^{2}\right)=\hat{e}\left(g_{1}^{\alpha W_{j}MN_{1}},g_{2}^{\beta N_{1}^{-1}A_{p}T_{P}^{⊤}}\right) \end{array}$$
and
$$\begin{array}{cc} E_{2} & =\hat{e}\left(C_{W_{j}}^{3},C_{T_{P}}^{3}\right)=\hat{e}\left(g_{1}^{\alpha W_{j}MN_{2}},g_{2}^{\beta N_{2}^{-1}B_{p}T_{P}^{⊤}}\right) \end{array}$$

Subsequently, the algorithm checks for the existence of its output $z_{j}\in S$ satisfying ${\left(D_{1}\right)}^{z_{j}}=D_{2}$, where *S* is a polynomial-sized subset of $Z_{q}$, $z_{j}=r\times d_{j}\left(T_{P}\right)+b_{P}$ and $d_{j}\left(T_{P}\right)=w_{1}t_{1}+\cdots +w_{m}t_{m}+b$. If no such $z_{j}$ exists, the algorithm outputs ⊥. Note that this algorithm demonstrates efficiency as |S|=poly(λ). Figure 5 illustrates the structure of vectors used in computing the classification result.

**Theorem** **1.***For each SVM model,* CS *uses the encrypted patients’ data to calculate the masked classification result.*

**Proof.** (5)$$D_{1}=\hat{e}\left(g_{1}^{\alpha },g_{2}^{\beta }\right)=\hat{e}{\left(g_{1},g_{2}\right)}^{\alpha \beta }$$(6)$$\begin{array}{cc} E_{1} & \;=\hat{e}\left(g_{1}^{\alpha W_{j}MN_{1}},g_{2}^{\beta N_{1}^{-1}A_{p}T_{P}^{⊤}}\right)\\ & =\hat{e}{\left(g_{1},g_{2}\right)}^{\alpha \beta W_{j}MN_{1}N_{1}^{-1}A_{p}T_{P}^{⊤}}\\ & \;=\hat{e}{\left(g_{1},g_{2}\right)}^{\alpha \beta W_{j}MA_{p}T_{P}^{⊤}} \end{array}$$similarly,
(7)$$\begin{array}{c} E_{2}=\hat{e}\left(g_{1}^{\alpha W_{j}MN_{2}},g_{2}^{\beta N_{2}^{-1}B_{p}T_{P}^{⊤}}\right)\\ =\hat{e}{\left(g_{1},g_{2}\right)}^{\alpha \beta W_{j}MB_{p}T_{P}^{⊤}} \end{array}$$
*Then,*

(8)
$$\begin{array}{cc} \; & D_{2}={\left(E_{1}\times E_{2}\right)}^{r}\\ & =\hat{e}{\left(g_{1},g_{2}\right)}^{r\alpha \beta W_{j}MA_{p}T_{P}^{⊤}+r\alpha \beta W_{j}MB_{p}T_{P}^{⊤}}\\ \; & =\hat{e}{\left(g_{1},g_{2}\right)}^{\alpha \beta rW_{j}M\left(A_{p}T_{P}^{⊤}+B_{p}T_{P}^{⊤}\right)}\\ \; & =\hat{e}{\left(g_{1},g_{2}\right)}^{\alpha \beta rW_{j}MM^{-1}T_{P}^{⊤}}\\ \; & =\hat{e}{\left(g_{1},g_{2}\right)}^{\alpha \beta rW_{j}T_{P}^{⊤}}\\ \; & =\hat{e}{\left(g_{1},g_{2}\right)}^{\alpha \beta r\left(MV_{j}\cdot DV_{P}\right)+b_{P}} \end{array}$$

Therefore, if $z_{j}=r\times d_{j}\left(T_{P}\right)+b_{P}\in S$, such that ${\left(D_{1}\right)}^{z_{j}}=D_{2}$, the secure classification algorithm outputs the masked classification result $z_{j}$. □

Finally, the *CS* transmits all the multiclass SVM model’s masked results ${\left\{z_{j}\right\}}_{j=1\cdots k}$ to P. To obtain the unmasked scores, the patient can subtract $b_{P}$ from ${\left\{z_{j}\right\}}_{j=1\cdots k}$. Then, the final classification result is the highest positive unmasked score.

## 5. Security Evaluations

This section uses a preposition/proof format to present the main security/privacy features provided by our proposal.

**Proposition** **1.**
*Our proposal can resist attacks launched by external eavesdroppers.*


**Proof.** As discussed earlier in the threat model section, attackers can be external eavesdroppers who have the capability to capture all messages exchanged in our scheme and they analyze these messages looking for vulnerabilities to infer sensitive information or compute the parameters of the diagnosis model. As explained earlier, the messages exchanged in our system are as follows: (1) the ciphertext of the diagnosis model sent from the model owner to the server; (2) the ciphertexts of the medical data sent by users to the server for classifications; and (3) the masked diagnosis returned by the server to the users. The first two types of messages are encrypted using private keys and without learning the keys, the external attackers cannot infer the messages. This is proved in the original paper of the inner product encryption we modified. For the diagnosis returned by the server to the users, as explained earlier, random numbers that are known only to the users are used to mask the diagnosis, and thus, because the attackers do not know these numbers, they cannot unmask the diagnosis to learn the diagnosis. □

**Proposition** **2.**
*Our proposal is robust against the known-ciphertext model attacks when the master secret key MSK is unknown to the attackers.*


**Proof.** As explained earlier, the matrices $A_{P},B_{p},N_{1}^{-1}$ and $N_{2}^{-1}$ are used to compute the secret key of the model users (${\mathrm{PSK}}_{P}$), and this key is used to compute the ciphertexts of the medical data. Additionally, the matrices $M,N_{1}$ and $N_{2}$ are used to create the secret key of the model owner (HSK), which is used to compute the ciphertext of the parameters of the diagnosis model. Our proposal is secure under the known-ciphertext model if the attacker does not know the master secret key MSK, which includes the matrices $N_{1}$, $N_{2}$, and *M*. The attackers can capture the ciphertexts of the medical data and the ciphertext of the model parameters, and without knowing the matrices $N_{1}$, $N_{2}$, and *M*, it cannot infer the medical data or the model parameters. □

**Proposition** **3.**
*Our proposal is secure under the known-background model, i.e., external attackers and the cloud server cannot compute the classification result (diagnosis).*


**Proof.** Under the known-background model, attackers are equipped with some background information, such as the distribution of diseases in a certain area to infer sensitive information. For instance, the work in [22] is not secure under the known-background model because the server can link the ciphertexts of the same diagnosis, so by using the distribution of a disease and the frequency of a ciphertext, the server can figure out the diagnosis of the users. In order to secure our proposal under the known-background model, we mask the classification result of the diagnosis model using random numbers that frequently change. These random numbers are known only by the users so they are the only ones who can unmask the results of the model and learn the diagnosis. In summary, external eavesdroppers and the server cannot learn the diagnosis because they are masked by a secret random number and cannot use background information to deduce the diagnosis because the random number frequently changes. □

**Proposition** **4.**
*Our proposal can resist the known-plaintext model attacks when the attackers do not know the users’ random numbers utilized to mask the diagnosis model’s output.*


**Proof.** In the *Known-plaintext model*, attackers can gather pairs of plaintext and corresponding ciphertext messages exchanged in the system and then analyze these pairs to infer any information that can be used to figure out new patients’ medical data. Because our proposal uses inner product operations to evaluate the diagnosis model, one way to launch the Known-plaintext model attack is by creating equations and solving them. Specifically, if each vector (either the medical data or the model) has $n_{e}$ elements, the attacker needs to create $n_{e}$ equations to solve them and obtain the $n_{e}$ unknowns. To do that, the attacker needs to learn $n_{e}$ medical data vectors and the result of the classification to make $n_{e}$ equations and solve them to compute the model’s parameters. The attack can also be launched with the objective of computing the medical data of the users. This attack can be launched by external eavesdroppers and/or the cloud server. In order to secure our proposal against this attack, we allocate the (m+2)th element in the medical data vector for a random number that should change every time the user sends data for classification. By doing that, we introduce a new variable each time the inner product is computed, i.e., the result of the inner product of the medical data and the diagnosis model is masked by the random number and is not known to the attackers. In summary, our proposal is secure against Known-plaintext model attacks because the attackers (either the cloud server or the data users) know only masked classifications (the result of the inner product of two vectors), and by changing the random number in each medical data, attackers cannot make equations and solve them to infer sensitive data because this random number adds a new variable in each equation. □

**Proposition** **5.**
*To preserve privacy, the same medical data (or same classification) of a patient sent at different diagnosis occasions are not linkable in our proposed approach.*


**Proof.** Even if attackers are not able to decrypt the ciphertexts in our system, sensitive information can be inferred if ciphertexts of the same data are linkable. For instance, if a user sends the same medical data on different occasions and attackers can link the ciphertexts, side information can be inferred such as no change in the users’ symptoms and vital data and thus no change in the health condition. Similarly, even if the attacker cannot unmask the masked diagnosis score that the server returns, linking the masked classifications of the same diagnosis reveals side information, such as no change in the health status of the users. To secure our proposed approach against this attack, in the encryption of the medical data, a random number (β) is used so that the ciphertexts of the same data look different when it is encrypted multiple times on different occasions. Additionally, by masking the output scores of the diagnosis model by different random numbers, the masked diagnosis scores of the same classifications look different. □

**Proposition** **6.**
*Our proposed approach can protect the confidentiality of the diagnosis model’s parameters.*


Prepositions 4 explains how the cloud server cannot compute the diagnosis model’s parameters leading to the protection of the intellectual property. Moreover, a random number (*r*) is used by the cloud server to prevent the users from computing the parameters of the model. Without using this random number, the users can compute the model’s parameters because they can use the medical data vectors and the diagnosis scores returned by the server to create equations and solve them to obtain the parameters of the model. Specifically, if the size of the vectors is $n_{e}$ elements, then attackers need $n_{e}$ equations by requesting $n_{e}$ diagnosis. Note that the diagnosis score is the inner product result of the vectors of the medical data and the model parameters. To protect our proposed scheme against this attack, a random number *r* is used by the cloud server to add a masking level to the output score resulting from the evaluation of the model. In this way, the user does not know the exact result of the product of the two vectors of the medical data and the model’s parameters and for each diagnosis operation, the random number *r* adds a new variable in the equations the user tries to solve. The server needs to change the random number each time it performs a diagnosis process. Note that the classification is determined by the sign of the classification score after unmasking the classification received by the users. This necessitates that the random number (*r*) is positive to avoid changing the sign of the classification value.

## 6. Experiments and Results

The purpose of this section is to quantitatively compare our proposal in this paper to the most relevant approaches in the literature. The section first presents the environment of the experiments and then discusses the results obtained.

### 6.1. Environment of the Experiments

In our experiments, to compute the computation and communication costs of our proposed approach, the Python charm cryptography library [36] is used. We used a computer with an Intel processor with Core i7-8700, a ram with 8 GB, and a frequency of 3.20 GHz. We implemented our proposal, in addition to the most relevant works in the literature, including the work done by Xie et al. [13] and the work done by Zhang et al. All results are presented in an average of 1000 trials.

Additionally, in order to evaluate the accuracy of the diagnosis model, we used the dataset in [19] that has data samples for six skin diseases to train our multi-class SVM model. These six diseases include *Psoriasis*, *Seboreic dermatitis*, *Lichen planus*, *Pityriasis rosea*, *Chronic dermatitis*, and *Pityriasis rubra pilaris*. The dataset has 366 data samples in total for six diseases, including 112 samples for *Psoriasis*, 61 samples for *Seboreic dermatitis*, 72 samples for *Lichen planus*, 49 samples for *Pityriasis rosea*, 52 samples for *Chronic dermatitis*, and 20 samples for *Pityriasis rubra pilaris*. Each data sample has 34 features. Examples of these features include scalp involvement, age, erythema, family history, polygonal papules, and mucosal involvement. While one feature takes a binary value, the remaining 33 features take numerical values.

In our experiments, we measure three metrics to evaluate our proposal and the most relevant ones, including *computation cost*, *communication cost*, and *diagnosis accuracy*. The *computation cost* measures the computation resources needed from each entity in our proposal, including model owners, users, and cloud servers. It is preferable to reduce this cost, especially on the users because they usually use resource-constraint devices like tablets and cell phones. The *communication cost* measures the amount of data (in bytes) exchanged between the different parties in our system. It is preferable to reduce this cost to reduce the amount of bandwidth needed in our system. The *diagnosis accuracy* measures the performance of the SVM model in two cases, including: (1) without protecting the model’s intellectual property and preserving user privacy, i.e., executing the model using plaintext data; and (2) using our inner product encryption cryptosystem, i.e., executing the model over encrypted data. The *diagnosis accuracy* is measured in terms of *Accuracy*, *False Alarm*, *Recall*, *F*1-*score*, and *Precision*, which are defined as follows.

*Accuracy* is the number of data samples that are diagnosed correctly to the total number of samples. The diagnosis model performs better as this metric increases. This metric is computed using this equation:
$$Accuracy=\frac{Tr_{p}+Tr_{n}}{Tr_{p}+Tr_{n}+Fa_{p}+Fa_{n}}\times 100,$$
where $Tr_{p}$ measures the number of data samples of sick people that are diagnosed correctly by the model, $Tr_{n}$ measures the number of data samples of healthy people that are diagnosed correctly by the model, $Fa_{p}$ measures the number of data samples of healthy people that are diagnosed incorrectly by the model, $Fa_{n}$ measures the number of data samples of sick people that are diagnosed incorrectly by the model.*False Alarm* is the number of data samples of sick people that are diagnosed incorrectly to the total number of samples of sick people. The diagnosis model performs better as this metric decreases. This metric is measured as follows:
$$FalseAlarm=\frac{Fa_{p}}{Fa_{p}+Tr_{n}}\times 100$$*Recall* is the number of samples of sick people that are correctly diagnosed to the total number of samples of sick people. The diagnosis model performs better as this metric increases. This metric is measured as follows:
$$Recall=\frac{Tr_{p}}{Tr_{p}+Fa_{n}}\times 100$$*Precision* is the number of samples of sick people that are diagnosed correctly to the total number of samples of sick people. The diagnosis model performs well as this metric increases. This metric is measured as follows:
$$Precision=\frac{Tr_{p}}{Tr_{p}+Fa_{p}}\times 100$$*F*1-*score* gives the average of Precision and Recall. This metric is especially a good indicator of the model’s performance when the dataset is not balanced. This metric is measured as follows:
$$F1-score=\frac{2PRRC}{PR+RC}\times 100,$$

### 6.2. Experimental Results

#### 6.2.1. Computation Cost

The computation costs for model owners at different numbers of diseases (or classes) and numbers of model features for our proposal, as well as the works in [12,13], are depicted in Figure 6a and Figure 6d, respectively. Comparing our proposal to those in [12,13], the two figures illustrate that our proposal imposes the least computation cost on model owners. This improvement is due to the fact that, unlike our proposal which only requires the encryption of the diagnosis model parameters, the proposal in [12] necessitates model owners’ participation in computing the diagnosis results in collaboration with the cloud server, in addition to the computations needed for model encryption. The figures also indicate that our proposal outperforms the work in [13] because it utilizes two non-colluding servers to execute the scheme. Thus, it requires the encryption of the diagnosis model twice using two different keys and distributes one ciphertext to each server.

The computation cost on the model users at different numbers of diseases (or classes) and numbers of model features for our proposal and the works in [12,13] are shown in Figure 6b and Figure 6e, respectively. Compared to the proposals in [12,13], the two figures show that our proposal imposes the least computation cost on the model users. The reason for this big reduction in the computation cost can be attributed to the use of an efficient inner product encryption cryptosystem compared to the use of Pailler cryptography and multi-party computation. Additionally, Figure 6b shows that the computation cost in our proposal and the work in [13] are constant at the different numbers of classes. On the contrary, the computation cost in the proposal of [12] increases with the increase in the number of classes. This can be attributed to the fact that in the proposal of [12], in addition to encrypting its medical data, the model users are required to compute the ciphertext of the model’s score and decrypt the diagnosis class. Figure 6e illustrates that the computation cost in our proposal and the two other works increases with the number of features. This is due to the fact that model users need to encrypt their medical data, and the size of this data depends on the number of features.

The computation costs on the cloud server at different numbers of diseases (or classes) and numbers of model features for our proposal and the works in [12,13] are shown in Figure 6c and Figure 6e, respectively. Compared to the proposal in [13], the two figures show that our proposal requires much less computation cost on the cloud server. This is attributed to two facts: (1) the inner product encryption cryptosystem used in our proposal is much more efficient than the multi-party computations needed in [13]; and (2) due to using two servers in the case of [13], computation resources are required from the two servers and the figure displays the total computation cost on the two servers. For the proposal of [12], the figures show that our proposal outperforms it when the number of classes is less than five and the number of features is less than thirty. We also observe a slight increase in the computation cost of our proposal compared to the proposal in [12] when the number of classes is five or more and the number of features is thirty or more. This is because our proposal allocates more computations to the cloud to reduce the computations on the model users and owners. This is totally acceptable because the cloud server has much more computation resources than the model users and owners.

As discussed earlier, we have modified the inner product encryption cryptosystem proposed by Kim et al. [15] that is designed for a single-encryptor single-decryptor setting to make it efficient for the single-encryptor multiple-decryptor setting, which fits our application due to the existence of one model owner (i.e., decryptor) and multiple users (i.e., encryptor). To evaluate this modification, we have implemented our SVM model using the cryptosystem of [15] and the modified cryptosystem in this paper. Figure 7 presents the computation costs on the model owner at different numbers of patients (or users) for our cryptosystem and using the cryptosystem of Kim et al. [15]. The figure shows that our modified inner product encryption cryptosystem significantly reduces the computation cost on the model owners and it does not depend on the number of users (or patients). The reason for this can be attributed to the fact that by using the cryptosystem in [15], the model owner needs to conduct one encryption operation for each patient because it is designed for a single-encryptor single-decryptor setting; on the contrary, our cryptosystem needs only one encryption operation by the model owner and this encryption can be used by all users because it is designed for a single-encryptor multiple-decryptor setting. In conclusion, our cryptosystem can reduce the number of encryption operations by the model owner from *n* to only one, where *n* is the number of patients.

#### 6.2.2. Communication Cost

The communication costs on the model owner at differ numbers of diseases (or classes) for our proposal and the works in [12,13] are shown in Figure 8a. The figure shows that our proposal requires less communication cost than the approaches proposed in [12,13]. Specifically, in the approach of [12], the model owner needs to send three messages to the server to send the ciphertext of the model and it also needs to send messages during the computation of the diagnosis. As explained earlier, the model owner is involved in the computation of the diagnosis in the approach proposed in [12]. Unlike this approach, our proposal is not involved in the computation of the diagnosis and it needs to send only one message to the server containing the ciphertext of the model. On the other hand, due to using two servers in [13], the model owner needs to send messages containing the model’s parameters to two servers instead of only one server in our proposal.

The communication costs on the model users at different numbers of diseases (or classes) for our proposal and the works in [12,13] are shown in Figure 8b. Compared to the approach of [13], the figure shows that the communication cost of our approach is lower and it does not depend on the number of classes in our proposal and the one in [13]. On the other hand, the figure shows a slight advantage for the proposal of [12] over our proposal. This can be justified as follows. In our proposal, the users send encrypted medical data to the cloud server that sends a masked diagnosis result to the users that unmask it to compute the diagnosis, but in the approach of [12], the users compute the encryptions of the diagnosis scores of all classes using encrypted model parameters and return the ciphertexts to the cloud server that returns the ciphertext of the exact diagnosis (or classification). Although the approach of [12] reduces the communication overhead slightly compared to our proposal, the cost of this improvement is a big increase in the computation cost on the model users, as shown in Figure 6b. This is not acceptable practically because the model users usually use resource-constrained devices such as tablets and cellphones.

The communication costs on the cloud server at different numbers of diseases (or classes) for our proposal and the works in [12,13] are shown in Figure 8c. The figure shows that the communication cost on the cloud server in our proposal slightly increases with the increase in the number of classes and is significantly lower than the costs of [12,13]. This big reduction can be attributed to these facts: (1) Compared to the work of [12], our proposal requires the cloud server to send only the masked diagnosis to the users, while in [12], the server requires sending encrypted model parameters to the users and exchanging several messages with the model owner to compute the diagnosis result; and (2) because of using multi-part computations in the approach of [13], it requires exchanging several messages between the two servers used in this approach.

#### 6.2.3. Diagnosis Accuracy

We have used the dermatology dataset in [19] to implement our diagnosis model and the models proposed by Zhang et al. [12] and Xie et al. [13]. We conducted two implementations: one for evaluating the model on plaintext data and the second one for evaluating the model over encrypted data (privacy preservation implementation). We have divided the dataset into two sets with a ratio of 8 to 2. The first set is used to train the SVM models while the second set is used to evaluate the models. To evaluate the diagnosis accuracy, we measured the metrics defined in Section 6.1, including *Accuracy*, *False Alarm*, *Recall*, *F1-score*, and *Precision*. The results we obtained are given in Table 1. The results indicate that the performance of our model is similar to that of [12] because both of them train linear SVM models. On the contrary, the approach in [13] trains a non-linear SVM model, which justifies its different performance results. The given performance results also indicate that the diagnosis accuracy is the same in the two implementations, which indicates that evaluating the models over encrypted data does not degrade the precision of the calculations compared to the evaluation using plaintext data. The table indicates that the *accuracy* in our model and the models of [12,13] are the same. However, the model of [13] exhibits a slightly better false alarm compared to our proposal and the model of [12]. This is because the model of [13] uses a non-linear SVM model that can create a more sophisticated decision boundary. This slight advantage of the model of [12] costs a significant increase in the communication and computation overheads as discussed in the last two subsections. This is because the execution of the non-linear operations of [12] over encrypted data requires extensive overhead, while the linear operations in our model and in [13] can be executed more efficiently over encrypted data. As discussed earlier, the proposed approach is efficient because it uses an efficient inner-product encryption cryptosystem. We used a dataset as a case study, and by using a different dataset, the model parameters may change, but our cryptosystem can still be used to execute these models over the ciphertext domain for protecting the confidentiality of the input data and the model’s parameters.

## 7. Literature Review

Our objective is to protect the confidentiality of the users’ data, health status, and model’s parameters and enhance the computational and communication efficiency. Additionally, we aim to create a diagnosis system that needs a single cloud server and does not need to involve the model owner in the computation of the classification result. To the best of our knowledge, our scheme stands out as the only one capable of achieving all of them. We refer to Table 2 for a concise comparison between our approach and the most relevant state-of-art works. Furthermore, this section delves into more detailed explanations of other schemes and compares them to our proposal.

Due to its potential, machine learning has been widely used in different medical diagnoses [37,38,39]. Artificial neural networks, SVMs, and regression trees are used in [40] for diagnosing Parkinson’s disease in early stages, while in [41], linear SVM is used to diagnose Alzheimer’s disease. In another example, random forest, Naive Bayes (NB), and SVM are used in [42] to diagnose COVID-19. There is no doubt that machine learning will play a major role in the realization of smart healthcare systems. Nevertheless, in most of the existing schemes, including the aforementioned works, machine learning models are evaluated using plaintext data and the outsourcing of these models to a third party that provides cloud diagnosis service endangers the privacy of the users and the model’s parameters. In comparison to these schemes, our proposal in this paper can achieve the objective of protecting the confidentiality of the model’s parameters and the users’ data and health status, which cannot be accomplished by the aforementioned schemes.

The privacy problem in machine learning-based medical diagnosis can be divided into two parts, including training and testing. While our paper aims to preserve privacy during the testing phase, several works in the literature have studied privacy in the training phase such as [11,16,17,18]. These works aim to train SVM models without exposing the sensitive information of the dataset of the training. The approach proposed in [43] trains the model using an encrypted dataset instead of using plaintext data to avoid revealing sensitive information. In [44], different entities have datasets for drug formulas and the work investigates an approach to enable a central unit to compute a model trained on all the datasets without sharing the data or exposing sensitive information. The work in [45,46] has investigated the use of blockchain technology to create a privacy-preserving approach for training SVM models, where the data are owned by different parties that do not want to reveal them. To do that, each entity stores the ciphertexts of their data on the blockchain and then these ciphertexts can be used to train the SVM model without being able to compute the plaintext data. The work in [28] investigates the use of mimic learning techniques to train a machine learning model without revealing the datasets that are owned by different parties. Each data owner trains a model, called teacher, using its local dataset, and then uses it to label insensitive unlabeled data used to train the usable model, called student. This process actually transfers the learning from the teacher model to the student without exposing the sensitive data to preserve privacy. In contrast to these works, which primarily focus on training machine learning models without exposing sensitive information, our emphasis in this paper lies in executing a medical diagnosis model while protecting the confidentiality of the users and the model’s parameters and building an efficient scheme by using lightweight cryptosystems and ensuring that it does not need involving the model owner in computing the diagnosis (or classification) scores.

Few works have investigated the evaluation of medical diagnosis models with privacy preservation. In [47], homomorphic encryption cryptosystem [48] is used to develop a medical diagnosing approach that can be run by a distrusted third-party server that provides cloud service. In [49], homomorphic encryption is used to evaluate a single decision tree medical diagnosis without revealing sensitive information on the patient’s data. The work in [9] has investigated the use of homomorphic encryption to evaluate a multiclass SVM medical diagnosis system hosted by a cloud server. However, as explained in [10], this work suffers from several limitations, where the most prominent one is exposing sensitive information by the server. Compared to these schemes, our approach can protect the diagnosis model’s confidentiality and preserve the privacy of the users’ data and health status, and it does not need the participation of the MO to compute the diagnosis scores while maintaining low resources in term of communication and computation.

The approach of [12] is close to our proposal. It uses the Okamoto–Uchiyama (OU) cryptosystem [50] to build an online medical diagnosis system based on multi-class SVM. It is worth noting that the homomorphic operations are used in the OU cryptosystem to evaluate the model over encrypted data. Nevertheless, there are several shortcomings associated with the approach proposed in [12], which can be summarized as follows. First of all, under the known-background model, the proposed approach reveals the classification (diagnosis) results to the cloud server, which creates a serious privacy issue. Specifically, because the server can link encrypted classifications that are for the same diagnosis, the server can easily determine these diagnoses by using some contextual data, like the prevalence of a particular disease in a certain area. The second shortcoming is that the MOs participate in the proposed computation of each diagnosis score in [12]. By doing this, the advantage of outsourcing the diagnosis model to a third party that provides a cloud service is diminished because it has to be always online and allocate computation and communication resources for the medical diagnosis process. The third shortcoming is that the MO has to send some model parameters to all users, if it needs to update the diagnosis model run by the cloude server, i.e., to add more diseases or increase the accuracy. The reason for this is that the proposed approach requires that the users execute a part of the model. In another notable shortcoming, because the users need to compute a part of the model, as just explained, the proposed approach imposes high computation overhead on the users. This requirement may be difficult to achieve practically because some users may use resource-limited devices like tablets and cellphones. Compared to the proposed approach in [12], our approach can hide diagnosis results without the need for online and interactive model owners to compute the diagnosis scores, in addition to requiring low resources in terms of communication and computation.

In [13], the authors attempted to address the limitations of [12] discussed above. The work in [13] proposes a multi-class SVM-based diagnosis system using two cloud servers that are assumed non-colluding. Although this system was able to address several limitations such as protecting the confidentiality of the diagnosis model’s parameters and the classification results, and computing the classification without the involvement of the model owner, it suffers from several limitations. Much more communication and computation resources are needed especially as more diseases (or classes) are diagnosed by the model. Additionally, two servers are needed to execute the proposed approach in [13], and the servers should not be colluding to ensure the security of the approach. Such a requirement is costly compared to our approach which needs only one server and the requirement that the two servers are not colluding is hard to achieve. Finally, the proposed approach uses a non-linear SVM model, and evaluating this model in the ciphertext domain needs extensive computation and communication resources because it is not easy to compute non-linear operations using encrypted data. In some applications, employing non-linear SVMs might seem appealing due to making a more sophisticated decision boundary compared to linear SVM. However, this choice often adds complexity to the scheme’s implementation in the ciphertext domain to preserve privacy. Our experiments in this paper showed that linear SVM gives good accuracy and, at the same time, it is easier to implement in the ciphertext domain. Compared to the approach proposed in [13], our proposal in this paper develops a single-cloud system and offers more efficient computation and communication overheads.

## 8. Conclusions and Future Work

We have developed a new privacy-preserving approach for SVM-based medical diagnosis. In our approach, the model owner encrypts the model’s parameters using an efficient cryptosystem that can compute inner product operations over encrypted data and then it sends the ciphertext to a cloud server run by a third party that provides cloud services. Additionally, users encrypt medical data, including symptoms and vital data, and send it to the server for diagnosis. The server can use the encrypted model and data to compute an encrypted classification without being able to figure out the plaintext model’s parameters and the users’ data. Our approach is based on an efficient inner product encryption scheme. This paper has modified a cryptosystem that can conduct inner product operations over encrypted data to make it more suitable for the setting considered in this paper, which includes one party (i.e., the model owner) that encrypts a vector (i.e., model parameters) while each user encrypts its medical data vector with a unique key while the inner product can be computed by the cloud server using these two ciphertexts. Moreover, unlike some existing proposals that require two non-colluding servers, our proposed approach is executed by only one cloud server. For the efficient implementation of the diagnosis system, all the classification computations are moved to the server without the need to involve the users or the model owner. To evaluate our proposal, extensive analyses and experiments are conducted, and the classification accuracy and computation and computation overhead are the main metrics measured. The results of our analysis demonstrate that our proposed scheme successfully fulfills our security and privacy objectives, including protecting the confidentiality of the user’s medical data and the diagnosis model parameters. Our experiments have indicated that our proposal consumes fewer resources in terms of computation and communication compared to the most relevant works in the literature while maintaining high classification accuracy.

## Figures and Tables

**Figure 1 sensors-23-09033-f001:**
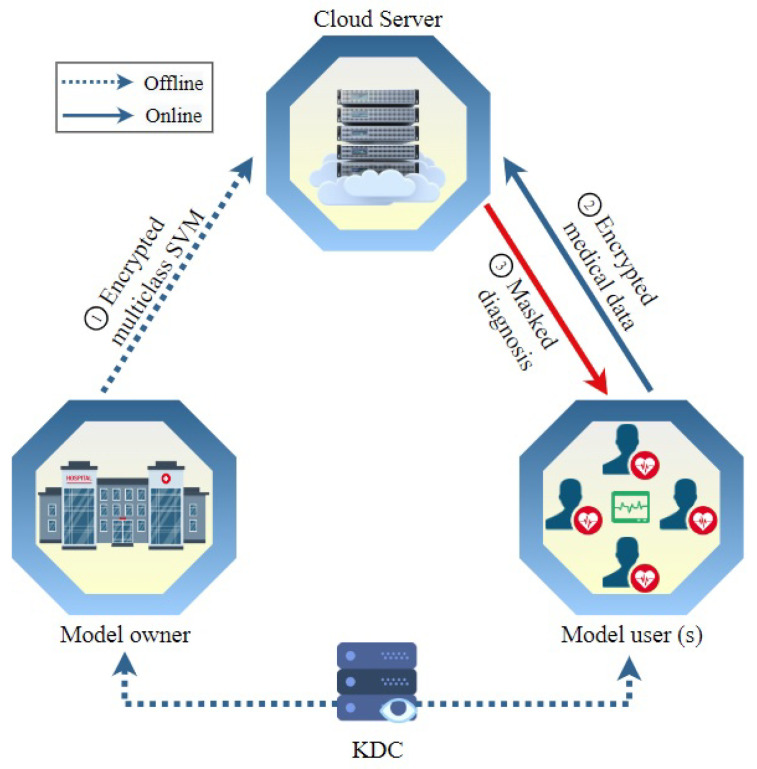
The network model considered in this paper.

**Figure 2 sensors-23-09033-f002:**
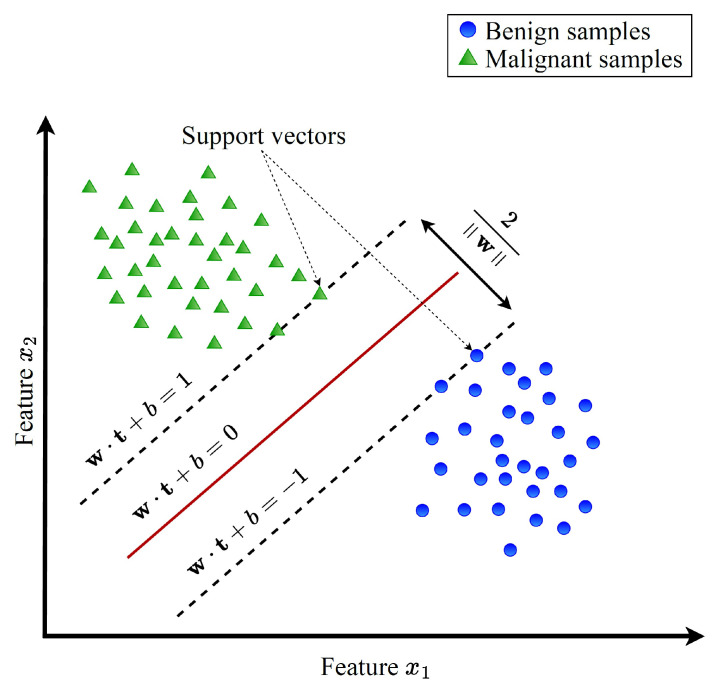
An illustration for linear support vector machine.

**Figure 3 sensors-23-09033-f003:**
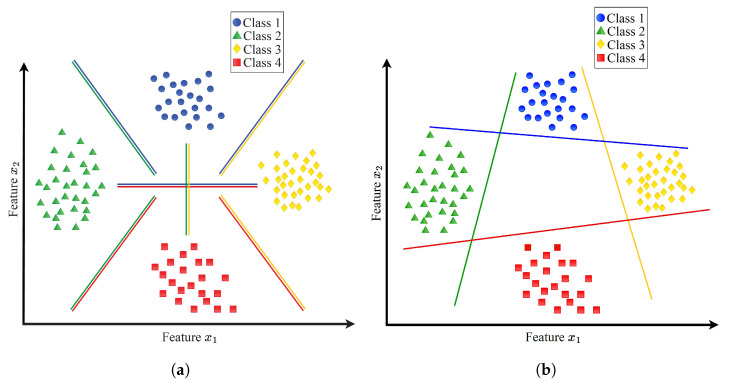
Multi-class support vector machine. (**a**) One versus one (1V1) multi-class support vector machine. (**b**) One-versus-all (1VA) multi-class support vector machine.

**Figure 4 sensors-23-09033-f004:**
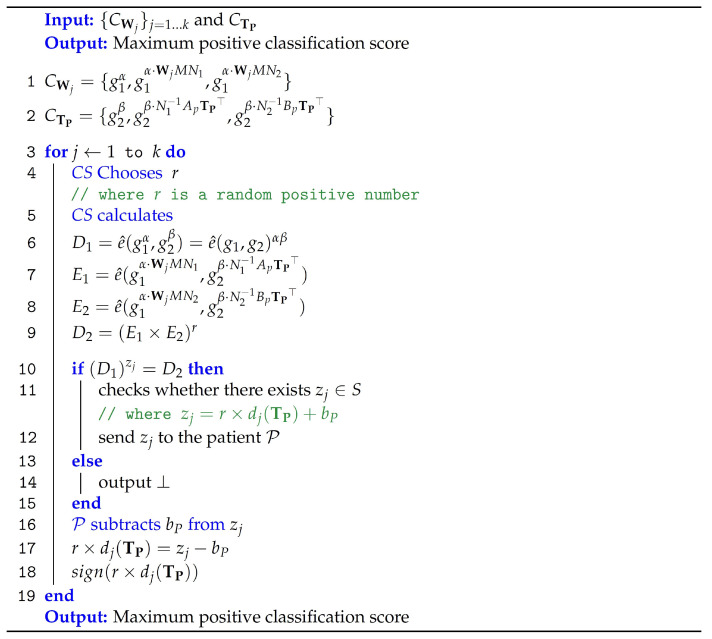
Medical diagnosis.

**Figure 5 sensors-23-09033-f005:**
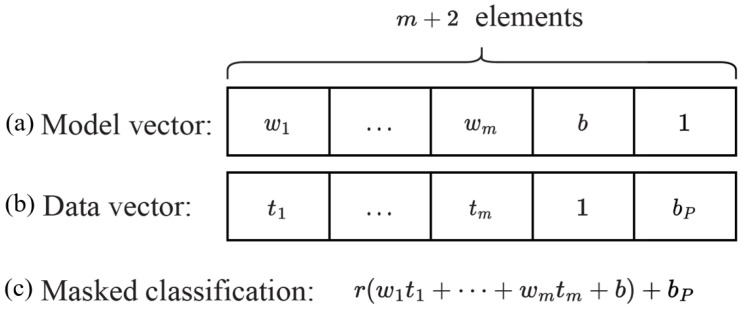
An illustration of the vectors utilized to compute the classification result is provided below. (**a**) Model vector: m+1 elements for the model parameters and the (m+2)-th element contains one. (**b**) Data vector: *m* elements for the patient’s medical data, one in the (m+1)-th element, and the (m+2)-th element comprises a random number $b_{P}$ utilized for masking the classification result. (**c**) The masked classification: calculated by the cloud server.

**Figure 6 sensors-23-09033-f006:**
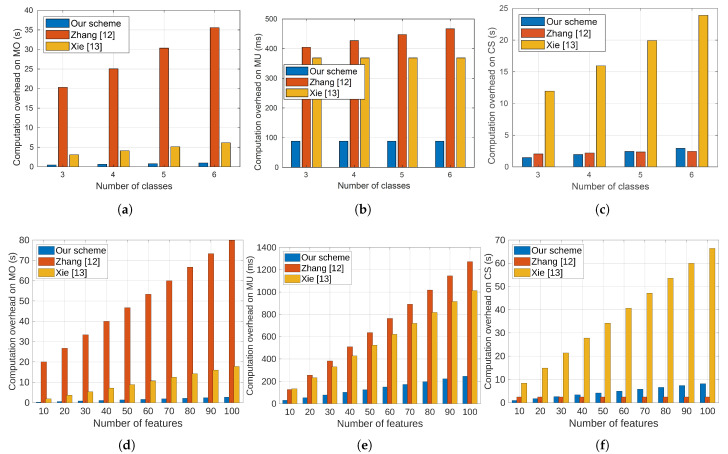
The required computation time by our approach and the most relevant ones varying the number of classes and the number of features. (**a**) Time on model owner vs. number of classes. (**b**) Time on model users vs. number of classes. (**c**) Time on cloud server vs. number of classes. (**d**) Time on model owner vs. number of features. (**e**) Time on model users vs. number of features. (**f**) Time on cloud server vs. number of features.

**Figure 7 sensors-23-09033-f007:**
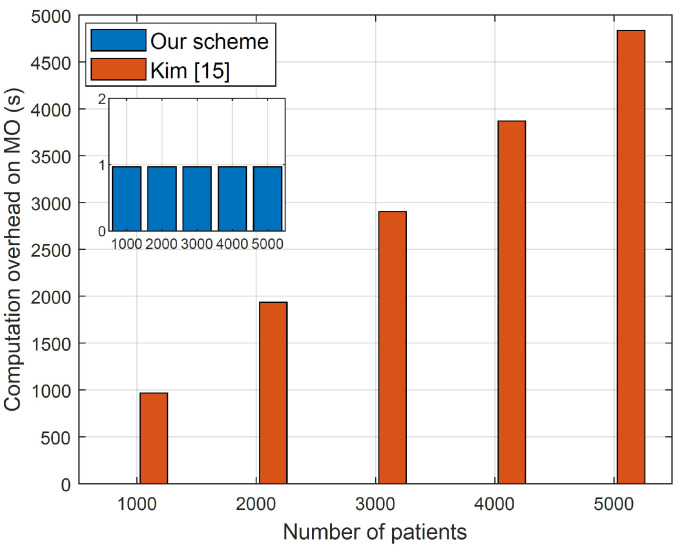
The computation time required by the model owner in the case of the original inner product cryptosystem proposed in [15] and our modified cryptosystem in this paper.

**Figure 8 sensors-23-09033-f008:**
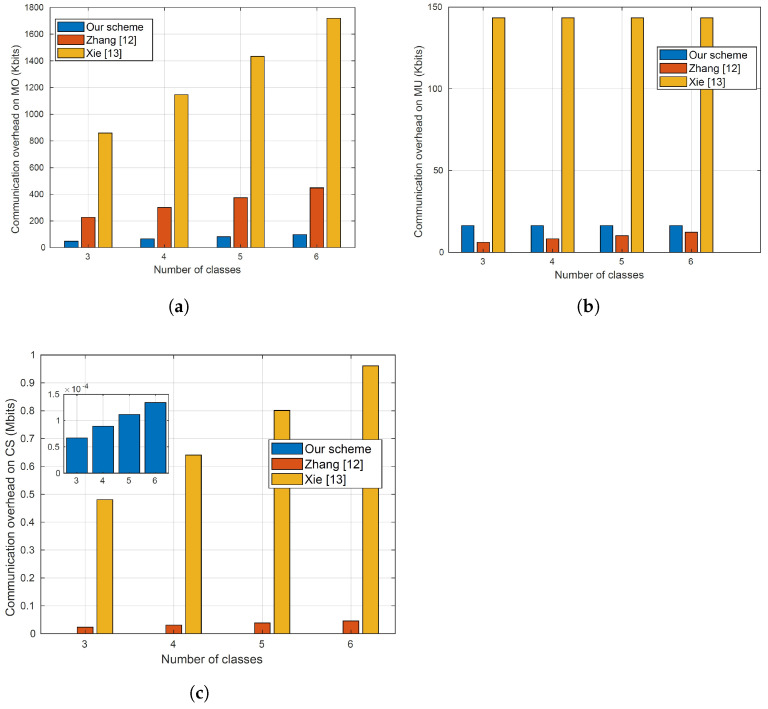
The communication costs of our proposal and the relevant works in [12,13]. (**a**) The communication cost on the model owner. (**b**) The communication cost on the model users. (**c**) The communication cost on the cloud server.

**Table 1 sensors-23-09033-t001:** Summary of the performance of our proposal and the most relevant works in the literature.

Metric/Scheme	[13]	[12]	Our Proposal
Accuracy	97%	97%	97%
Precision	97%	97%	97%
Recall	98%	98%	98%
F1-score	97%	97%	97%
False Alarm	0.4%	0.5%	0.5%

**Table 2 sensors-23-09033-t002:** Comparison between our proposal and the state-of-art works.

	[8]	[9,10]	[12]	[13]	Our Scheme
Uses cloud server	*√*	*√*	*√*	*√*	*√*
Preserving the privacy of patients	*√*	*√*	*√*	*√*	*√*
Model’s intellectual property protection	×	×	*√*	*√*	*√*
Concealing classifications	*√*	*√*	×	*√*	*√*
Multiple classes	×	*√*	*√*	*√*	*√*
Uses a single cloud	×	*√*	*√*	×	*√*
Computation of diagnosis without the MO	×	×	×	*√*	*√*
Requires low resources	×	×	×	×	*√*

## Data Availability

Not applicable.

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
