# Peer review of "Lightweight Multi-Class Support Vector Machine-Based Medical Diagnosis System with Privacy Preservation"

_sensors, 2023, doi:10.3390/s23229033_

Round 1

Reviewer 1 Report

The paper presents a novel scheme for privacy-preserving medical diagnosis using multi-class Support Vector Machines (SVM) in a cloud-based environment. The proposed scheme aims to protect patient privacy, preserve the model's intellectual property, and reduce computational overhead. It employs a function-hiding inner product encryption technique, which is more efficient than existing cryptographic methods, to enable secure data classification. The paper also reviews related works and highlights the limitations of previous approaches. Overall, the paper offers an innovative solution to the challenges of privacy and efficiency in medical diagnosis while providing a comprehensive comparison with existing methods.

The abstract of the paper effectively highlights the proposed scheme's security and reduced overhead compared to existing schemes. However, it could benefit from a brief discussion of the potential real-world implications or applications of these findings. Such a discussion would help readers understand the practical relevance of the research and its potential impact on various domains.

The introduction provides a comprehensive overview of the context and challenges related to telemedicine and privacy-preserving medical diagnosis. However, it lacks explicit identification of the research gap that the proposed scheme aims to address. Clearly stating the specific problems and limitations in existing schemes that your research seeks to solve would provide readers with a better understanding of the motivation behind your work.

The introduction mentions that the proposed scheme focuses on multi-class SVM for medical diagnosis. However, it could benefit from briefly explaining why multi-class SVM is important in the context of medical diagnosis and how it differs from binary classification. This clarification would help readers grasp the significance of your choice of methodology.

While the introduction mentions the use of a real dataset for training the model, it would be helpful to provide a clear motivation for the selection of this dataset. Explaining why this dataset was chosen and how it relates to the research's practical implications can enhance the reader's understanding of the study's relevance.

The introduction briefly mentions the main contributions of the paper at its end. To provide readers with a clearer roadmap, it would be beneficial to present a concise list of these contributions upfront. This would give readers a clear overview of what to expect from the paper.

The introduction mentions that the paper is derived from the first author's PhD thesis but does not explain how this research builds upon or extends the work presented in the thesis. A brief explanation of the relationship between the paper and the thesis could help readers understand the evolution of the research.

In the section that reviews related works and compares them to the proposed scheme, you have covered various important elements. To enhance the comprehensiveness and clarity of this review:

Consider providing a brief preview of the comparison table (Table 4) at the beginning of this section to help readers anticipate the upcoming detailed comparisons.

Explicitly state how your proposed scheme differs or improves upon each existing work after discussing them. Highlighting your scheme's unique contributions and advantages in each case will make the comparison more focused and meaningful.

Providing quantitative metrics or examples to illustrate the shortcomings of previous schemes, such as computation or communication overhead, would make these limitations more tangible for the reader.

When referring to cryptographic methods like Paillier cryptography and homomorphic encryption, consider providing a brief explanation or reference for readers unfamiliar with these terms to enhance their understanding.

Clarify why the choice of a non-linear SVM model in one of the schemes is relevant and how it impacts the scheme's performance compared to linear SVM models.

Discuss the scalability of each scheme and how they handle an increasing number of classes or data points, as scalability is often critical in real-world applications.

Emphasize how each scheme addresses privacy and security concerns, particularly regarding patient data and model intellectual property, to highlight the importance of your proposed scheme in addressing these issues effectively.

Addressing these points will improve the clarity and comprehensiveness of the review, helping readers better understand the significance of your proposed scheme within the context of existing research.

Reviewer 2 Report

The authors have enhanced an inner product encryption scheme in their medical system, where a single entity owns the model and multiple users are involved. This improvement enables the cloud server to conduct diagnoses like classification without exposing sensitive patient data or revealing model parameters. Crucially, it achieves this with minimal communication and computational overheads, utilizing only a single server. Analysis is provided to confirm the security and privacy preservation of their scheme, while empirical experiments are presented to reveal its superiority with reduced communication and computation overhead compared to existing approaches.

However, there are several points of concern regarding this paper:  

  1)The paper lacks clarity in comparing its approach with existing research. For example, it mentions that some existing research does not hide classification results. Consequently, it raises the question of whether, even when the result is concealed, the proposed approach remains more efficient. This query applies to other related works as well. Therefore, it is essential to clarify whether extending this work to encompass other existing methodologies still results in the proposed approach being more efficient. Additionally, the absence of a comparison with existing methods for two-class scenarios should be addressed.     2) The emphasis on using a medical dataset is not well substantiated. It remains unclear whether employing a non-medical dataset, such as the adult dataset, would yield different outcomes or have any impact on the proposed approach's performance.    3)  A minor comment relates to the presentation of vectors in bold. For instance, in Formula (1), it is suggested that x_s is a vector, and it would be beneficial for clarity if it were presented in bold.     4) The figures in the paper are somewhat disconnected from the pages where their descriptions can be found. Improving the placement and alignment of figures with their corresponding descriptions would enhance the reader's experience.     5) There are concerns about the presentation of results in Figure 6 as a barplot. It appears that the bars have not been compared in a unified manner, and it is unclear why only one technique is compared. A more comprehensive analysis with appropriate comparisons should be included.   6) The order of presentation is a bit unusual, with the dataset description appearing after multiple experimental results related to the dataset. It would be more logical to introduce and describe the dataset before presenting experimental results involving it.    7) It is suggested that varying the number of features should be considered to assess its impact on computational cost. Therefore, it is advisable to include an experiment that varies the number of features and includes it in the comparison with other methodologies.     Addressing these concerns and making the necessary adjustments will enhance the clarity and completeness of the paper.
